# Exploratory and Confirmatory Factor Analysis of the Social Skills Scale for Young Immigrants

**María Tomé-Fernández** **, Christian Fernández-Leyva \*** **and Eva María Olmedo-Moreno**

Department of Research and Diagnostic Methods in Education, University of Granada, 18194 Granada, Spain; mariatf@ugr.es (M.T.-F.); emolmedo@ugr.es (E.M.O.-M.)
\* Correspondence: cristianoronaldo@correo.ugr.es; Tel.: +34-675-278-566

**Abstract:** The integration of young immigrants in the societies that host them highlights the need for the intervention of social workers to facilitate their adaptation and inclusion from an individualized diagnosis of their needs. The development of social skills in the immigrants is one of the main ways to make that integration happen, and therefore its diagnosis is fundamental. However, at present, there are no valid and reliable instruments that take into account the sociocultural factors that surround young immigrants for the evaluation of their social skills. It is for this reason that the purpose of this study was to adapt and validate a current and useful instrument for the diagnosis of such social skills to young immigrants welcomed in Spain. To do this, it was started on the choice and adaptation of The Social Skills Scale (Escala de Habilidades Sociales). Subsequently, the questionnaire was submitted to concurrent, predictive, and nomological validation processes. The construct validity was carried out by factor analysis first and second order to confirm the hierarchical structure of the scale. After validation with Exploratory Factor Analysis ($n = 330$), the structure was checked, and the model was later adjusted with Confirmatory Factor Analysis ($n = 568$) by means of structural equations. The reliability and internal consistency of the instrument was also tested with values in all dimensions above 0.8. It is concluded that this new instrument has 29 items and 6 dimensions, has acceptable validity and reliability, and can be used for the diagnosis of Social Skills in Young Immigrants.

**Keywords:** immigrants; instrument; social skills; validation; factor analysis; structural equations

## 1. Introduction

The assessment of social skills has been a difficult area of research since its inception [1,2]. This difficulty is emphasized if the diagnoses of ideas, feelings, and opinions [3] are made for young immigrants.

The social context of the immigrant in his adolescent and youthful stage is difficult [4]. When the child arrives in the receiving country, he faces high levels of unemployment, marginalization, social exclusion, and discrimination, which leads to a steep growth in the social gap towards him [5] and social skills, that he owned, are depleted [4]. Social skills are defined as behaviors through which people are able to express ideas, opinions, feelings, and affection for others [6].

More specifically in Spain, young immigrants face different challenges, which vary depending on the socioeconomic level and the country of origin [7]. For example, to young people, immigrants from Morocco, this being the population from which the most immigrants arrive in Spain, together with the Romanian population, are those who arouse the least sympathy among Spanish natives, which causes greater rejection towards them and consequently poor academic performance [8]. This translates into socioeconomic disadvantages, due to the lack of employment and a proper integration process, getting in some Spanish cities to be a segregated culture [4,5].

However, a large number of Latin American migrants arrive in Spain, attracted by the call effect and by the similarity in language [9]. The Latin culture, similar to the Spanish one, together with the language, facilitate their social integration, and this, in turn, is reflected in the socioeconomic level, because they find paid jobs in the country more easily than other groups [10].

In addition, immigrants from the European continent arrive in Spain, especially in search of certain sanitary conditions and the warm climate that characterizes the country [11,12]. These immigrants usually form their own communities, even going so far as to create schools linked to the educational system of the country of origin [13].

Finally, in recent decades, the number of immigrants of Asian origin in Spain has increased. The greatest difficulty for these are found in language [14]. As with the previous group, they limit social relations to people from the community, although this is disappearing in generations to come. This may be due to the limited time they spend interacting [15,16].

The difficult integration of young immigrants in host societies [17] emphasizes the need for action by social workers to facilitate their adaptation and real inclusion [18]. When arriving in a country with cultural and linguistic differences, young immigrants are seen, in some situations, as separated from the rest of their equals [3]. Because of this, they need to learn the customs and language of the receiving country, and, therefore, learn to communicate and socialize to achieve full social inclusion [2]. The development of social skills is a key element for this to happen, decreases social anxiety and fear of rejection [19], enhancing the confidence of immigrants and therefore increasing the number of interpersonal relationships between them and people of different cultural and linguistic origins [20].

Currently, the struggle to find a better future has increased the number of young immigrants in western cities, nurturing the social and educational institutions of adolescents of diverse cultures, ethnicities, and religions [21,22]. For example, in Spain, 10% of the total student population are immigrants, which means about 400,000 young people in school (Instituto Nacional de Estadística, 2017) who need the help of social workers for the social inclusion of these and their families [23]. The social integration of these young people is one of the greatest current challenges, although the strategies that have been put in place for this purpose still do not achieve the desired results, especially in relation to the development of social skills in educational institutions [2,24,25], which is the first context in which they need to socialize.

The problem is that obsolete instruments of self-report of social skills published in the 1970s are still used [2,26,27], added to the difficulty of finding culturally sensitive instruments [28] for young immigrants. What makes it difficult for social workers is their subsequent intervention, since the diagnosis of the social skills needs of this sample is not carried out effectively [29].

In addition, it is worth mentioning the lack of instruments for the diagnosis of social skills that were built taking into account the linguistic difficulties that may arise in culturally diverse participants [30–32]. Fact that bases the objective of this research, that adapt and validate a new instrument whose items were simple enough to be understood by those subjects who dominated the Spanish language in an incipient manner and also allowed the quick translation of these, for those young people who totally ignore the language. For this reason, the use of consolidated questionnaires was discarded in this investigation, such as the Social Skills Questionnaire (CHASO-III) [2], the Social Skills Questionnaire for College Students (SSQ-U) [33] or the Social Skills Inventory [34].

On the other hand, following theoretical indications [35,36], a short instrument was sought, which could be applied quickly and inexpensively, whose corrections were easy to make and interpret; the reason that supported the discard of instruments with a large number of items such as The Social Skills Improvement System-RS [26] or The Social Skills Questionnaire for Traumatic Brain Injury (SSQ-TBI) [27].

The use of questionnaires that did not fit the dimensions considered important for the analysis of social skills was also rejected [37–39], such as the Skills Learning Team Questionnaire (CHSEA) [40] or those whose content and expression did not fit the age and reading comprehension of the selected

sample; for example, the Social Competence in Higher Education Questionnaire (CCSES) [41] or The Social-Emotional Assessment/Evaluation Measure (SEAM) [42].

Choosing for the adaptation and validation on which this research is based, the instrument that, complying with the aforementioned requirements, verifies the best psychometric properties of validity and reliability [43], in addition to being current for the treatment of social skills, is based on theoretical models of robust behavior [27], and it is based on the pattern of the Social Skills Scale (EHS) [1].

The instrument is composed of six underlying constructs. The first of these, the ability to say no and cut interactions, pretends to know the ability of young people to refuse to perform a task when they think it is not fair for them. To say no, it is a way of expressing social skills and defending a person's right to choose what is right [39].

The second construct, self-expression in social situations, aims to find out the degree to which young immigrants are capable of carrying out activities of daily life. For this, the trust they have towards themselves is analyzed, since one of the indicators of self-expression in social situations is the confidence in their own possibilities [38].

With respect to the third construct of the instrument, defense of one's rights as a consumer, it is used to detect the immigrant's ability to demonstrate his knowledge of his own rights when he goes to a public establishment. In relation to this, it highlights the right to security, the right to be informed, the right to be heard, the right to choose, the right to privacy, and the right to compensation as fundamental to the social development of a subject [37].

With the fourth construct, expressing anger or disagreement, the questionnaire identifies the ability to externalize anger towards other people. This anger is defined as a negative emotional response to the blocking of the objective and the unjust behavior of others towards oneself, and they affirm that this construct characterizes the social skills that are used as a shield against harm caused by other people [44].

The fifth construct is the ability to make requests, through which it is intended to measure the degree to which young immigrants are able to ask for something they consider their own. The ability to petition causes the individual to leave his internal forum and externalize a problem that he does not know or cannot solve by himself [45]. In this way, the subject must relate to others, leaving shyness aside.

The sixth construct is identified with the factor initiating positive interactions with the opposite sex, through which it is intended to know if the participants are able to hold conversations and interact with people of the opposite sex. This ability is related to spontaneously making a compliment or simply being able to talk to someone who is attractive [46]. It is important to note that interactions with the opposite sex become increasingly common and important in adolescence and that, although they are exciting, these interactions are full of ambiguity due to the relative lack of romantic experience and to the lack of clarity about the norms that regulate them [47].

## 2. Materials and Methods

### 2.1. Instrument Development

In a first phase, a review of the relevant literature and the exploration of instruments related to the topic was carried out, such as the Social Skills Learning Team Questionnaire (Cuestionario de habilidades sociales de equipos de aprendizaje) (CHSEA) [40], Social Skills Questionnaire (Cuestionario de habilidades sociales) (CHASO-III) [2], the Social Skills Inventory [34], the Social Skills Questionnaire for College Students (SSQ-U) [33], the Social Skills Improvement System-RS [26], the Social Skills Questionnaire for Traumatic Brain Injury (SSQ-TBI) [27], the Social-Emotional Assessment/Evaluation Measure (SEAM) [42], the Social competence in Higher Education Questionnaire (Cuestionario de competencias sociales en Educación Superior) (CCSES) [41], or EHS [1]. Subsequently, the research group decided to adapt the last-mentioned scale to the sample of immigrants, considering it more appropriate in terms of the simplicity of the language of the constructs evaluated, in addition to

complying with excellent psychometric properties of validity and reliability [44] and of being novel in the definition and diagnosis of social skills; remaining, an initial questionnaire of 37 items.

Subsequently, a first validation of the questionnaire was carried out through the expert judgment technique and with the participation of 13 professionals. Seven of the experts were professionals related to the development and social skills of young people. These worked in non-Governmental Organization, educational institutions and public posts linked to intervention activities to improve emotional and social relationships. The remaining six were researchers from different Spanish universities, whose lines of work were linked to the diagnosis and evaluation of social aspects in immigrants.

The initial scale presented to them was composed of 7 elements for the dimension that measures the ability to say no and cut interactions, 9 elements to measure self-expression in social situations, 7 elements to measure the defense that makes the rights themselves as a consumer, 5 elements for the factor that measures the expression of anger or disagreement, 6 elements to measure the ability to make requests to others, and 3 elements for the dimension that measures the ability to initiate positive interactions with people of the opposite sex. The experts established the content validity of this instrument through the content validity index for each element (I-CVI). More specifically, they assessed the relevance of each item to assess the social skills of young immigrants.

The questionnaire and the method of escalation were sent through email, which indicated that they value from 1 to 5 (1 = not relevant, 2 = somewhat relevant, 3 = relevant, 4 = quite relevant, and 5 = very relevant) the importance of each item. The elements that scored with I-CVI < 0.78 were extracted from the instrument [48], leaving the questionnaire with 33 elements. In addition, the experts evaluated the content validity index of the scale in general (S-CVI); this was done taking into account the average I-CVI of each item [49], obtaining a S-index CVI = 0.92, which reflected excellent content validity [48].

The apparent validity and readability of the questionnaire were also consulted upon with the experts, asking them to indicate qualitatively and answering open questions, the adequacy of the information of the instrument for immigrant young, as well as the understanding and writing of the items [50]. Thus, taking into account the experts' responses, several elements were modified in a nonsignificant way and 4 were substantially modified.

Finally, the final questionnaire was obtained, demonstrating its hierarchical structure through the performance of factor analysis. For this, the quantitative data acquired through the application of the instrument to the participants in two periods were analyzed; a first period, through which was carried the Exploratory Factor Analysis (EFA) by means of the SPSS statistical software in its latest version, involved the elements of the instrument being reduced to 29 items organized in six dimensions (the ability to say no and cut interactions formed by 7 items, the self-expression in social situations formed by 8 items, the defense of one's rights as a consumer formed by 4 items, the expression of anger or disagreement formed by 3 items, the ability to make requests to others formed by 4 items, and the ability to initiate positive interactions with people of the opposite sex consisting of 3 items). In addition, this involved a second period, in which the Confirmatory Factor Analysis (CFA) was carried out through the AMOS statistical program, in where is ratified the hierarchical structure of the previous analysis. The 29 items were divided in the same way into the aforementioned dimensions.

## 2.2. Scoring of Scales

All the categories of the instrument are measured through a rating scale about frequency that ranges from never (1) to always (4). In this way, the ability to say no and cut interactions is measured by seven elements to achieve a possible range of 7 to 28. The self-expression in social situations of participants is measured with eight items to achieve a possible range of 8 to 32. The defense of one's rights as a consumer is measured by four items to achieve a possible range of 4 to 16. The expression of anger or disagreement is measured with three items to achieve a possible range of 3 to 12. The ability to make requests to others is measured by four items to achieve a range of 4 to 16, and the ability to initiate positive interactions with people of the opposite sex is measured by three elements to achieve a

range of 3 to 12. To obtain the scores that differentiate the presence or not of the ability to average the scores in each of the dimensions will be followed [51]. In this way, the ability to "say no and cut interactions" will be obtained when the score is less than 17.5 points, the ability of "self-expression in social situations" with a score less than 20 points, the ability of "defense of one's rights as a consumer" with a score of less than 10 points, the ability to "express anger or disagreement" with a score of less than 7.5 points, the ability to "make requests" with a score of less than 10 points and the ability to "initiate positive interactions with the opposite sex" when the score is less than 7.5 points.

The instrument aims to measure social skills in young immigrants. For this, it has a series of distinctive characteristics that make it appropriate for the measurement in the context and the selected sample. These characteristics mainly focus on simple language, easy to read and understand in subjects in the process of learning the language, as well as viable and quick to translate in cases where it is necessary [52–54].

In addition, the language of the questionnaire is inclusive and generalizable [55–57], avoiding hurtful, racist, or intolerant words that cause a negative reaction to the answer of the items [58]. Also, this avoids value judgments or any expression that relates the item to the different ethnicities, cultures, or religions of the selected sample, so that the instrument allows generalized measurement of culturally diverse samples.

The items of the instrument represent simple actions of daily life that will allow to diagnose the existence or not of the underlying social ability, considering it essential for the improvement of the subsequent intervention, as the lack of such skills tends to be associated with low acceptance, rejection, low self-esteem, helplessness, maladjustment, delinquency, or addictions in young immigrants [59,60]. This is a fact that would hinder their social integration and their necessary acculturation [61–63].

### 2.3. Pilot Test

After the Institutional Review Board of the University of Granada approved the study, the instrument was applied in a pilot test to 20 young immigrants from the city of Granada, with similar characteristics to the study target population, to determine readability, the time of completion, and the understanding. The results indicated that adolescent immigrants between 12 and 18 years old could understand and perform the questionnaire successfully.

### 2.4. Sample and Procedure

For both factor analyses, the sample was selected through a nonprobabilistic sampling for convenience, in which the participants were the young immigrants belonging to the social and educational institutions of different Spanish cities who wanted to participate.

Table 1 shows the number of young people that make up the study sample in each city. In all of them, the confidentiality of the data was assured, and the questionnaire was administered anonymously.

**Table 1.** Number of young immigrants who participated in the study in each city.

| Province | Immigrants in Exploratory Factor Analysis | Immigrants in Confirmatory Factor Analysis |
|---|---|---|
| Granada | 77 | 206 |
| Málaga | 50 | 51 |
| Almería | 47 | 69 |
| Jaén | 16 | 27 |
| Córdoba | 18 | 22 |
| Cádiz | 40 | 62 |
| Sevilla | 12 | 27 |
| Huelva | 33 | 49 |
| Ceuta | 18 | 27 |
| Melilla | 19 | 28 |
| Total | 330 | 568 |

The questionnaire was completed by a total of 330 immigrants for the EFA. The demographic data that the EFA showed were that 54.8% were men and 45.2% were women, with an average of X = 14.82 years (SD = 1.7). The countries of origin of the participants included Morocco (33.9%), Bolivia (10%), Ecuador (5.2%), China, Colombia, and Japan (3.9%), Venezuela (3.6%), Argentina (3.3%), Paraguay (3%), Senegal (2.7%), Brazil (2.1%), Uruguay, Nigeria, Ukraine, and Algeria (1.5%), Cote d'Ivoire, Mali, South Korea, Russia, Italy and France (1.2%), Pakistan, Dominican Republic, Cuba, Romania, and England (0.9%), Honduras, Mexico, Peru, Syria and Germany (0.6%), and Nicaragua, Guinea, Kosovo, Greece, Latvia, Sweden, Poland, Bulgaria, Switzerland, Portugal, and Georgia (0.3%).

For the CFA, the questionnaire was completed by a total of 568 immigrants; 56% were men and 44% women X = 13.66 years (SD = 1.4). The countries of origin of the participants included Morocco (43.7%), Ecuador (9.2%), Romania (6.7%), Senegal (3.7%), Peru (3.3%), China and Colombia (3.2%), Bolivia (2.6%), Japan (2.5%), Mexico, Argentina, and Brazil (2.1%), Venezuela (1.9%), Portugal (1.8%), Germany (1.6%), Italy and Cote d'Ivoire (1.4%), Nigeria (1.2%), Chile (1.1%), France (0.9%), Mali (0.7%), Honduras, Cuba, and England (0.5%), Bulgaria, Russia, Algeria, and Syria (0.4%), and the Dominican Republic, Uruguay, Guinea, and Poland (0.2%).

### 2.5. Reliability Tests

The reliability of the final instrument was made using the test–retest reliability and internal consistency method in order to test and retest the reliability of the questionnaire. The instrument was administered twice at two-week intervals to 30 participants. As the theorists [64] show, acceptable levels were obtained for Cronbach's alphas. It is considered excellent when Cronbach's alphas is over 0.70 [65] (Table 2).

**Table 2.** Cronbach's alphas and correlation coefficient between the subscales.

| Cronbach's Alphas | Correlation Coefficient between the Subscales |
|---|---|
| Test 0.775 | 0.105 |
| Re-test 0.826 | 0.105 |

### 2.6. Criterion Validity

In order to know if the adapted instrument correlates with the already measured elements of the original instrument, the concurrent and the predictive validity of the same was carried out utilizing a sample of 30 subjects.

To carry out the concurrent validity, following theoretical indications [66,67], we utilized the original questionnaire [1] and the final adapted questionnaire, correlating the variables of both instruments through Spearman's correlation coefficient [68], due to the non-normal distribution of the data, where it was obtained that all the evaluated elements correlate positively with Spearman's Rho over 0.87 (Table 3). This justifies an excellent concurrent validity of the Social Skills Scale for Young Immigrant.

**Table 3.** Spearman's rho test for concurrent validity.

| | Social Skills Scale for Young Immigrant | | | | | |
|---|---|---|---|---|---|---|
| | 1. Say No | 2. Self-Exp. | 3. Rights Def. | 4. Angry Exp. | 5. Make Requ. | 6. Int. Opp. Sex |
| 1. Say no | 0.95 | | | | | |
| 2. Self-Exp | | 0.91 | | | | |
| 3. Rights D. | | | 0.87 | | | |
| 4. AngryExp. | | | | 0.90 | | |
| 5. Make Req. | | | | | 0.98 | |
| 6. Int. Opp. S. | | | | | | 0.99 |

The predictive validity of the instrument was also carried out. For this, the original instrument [1] and the final instrument, with a three-month difference period, were applied to a sample of 30 people. In this case, Spearman's correlation coefficients were greater than 0.80 (Table 4), which evidences an excellent predictive validity of the SSYI.

**Table 4.** Spearman's rho test for predictive validity.

| | Social Skills Scale for Young Immigrant | | | | | |
|---|---|---|---|---|---|---|
| | **1. Say No** | **2. Self-Exp.** | **3. Rights Def.** | **4. Angry Exp.** | **5. Make Requ.** | **6. Int. Opp. S.** |
| 1. Say no | 0.85 | | | | | |
| 2. Self-Exp | | 0.80 | | | | |
| 3. RightsD. | | | 0.80 | | | |
| 4. Angry Ex. | | | | 0.81 | | |
| 5. Make Re. | | | | | 0.90 | |
| 6. Int.Opp.S. | | | | | | 0.93 |

*2.7. Nomological Validity*

In order to show the nomological validity of the SSYI, the means of the scores, and the respective standard deviation (SD), were compared between the group of immigrant women and men, using the variation coefficient [69]. According to theoretical approaches, women were expected to present greater social skills [70–72]. Results were obtained that ratify the underlying theory, which was evidenced by a good nomological validity for the instrument (Table 5).

**Table 5.** Coefficient of variation for nomological validity.

| Dimension | Men | Women |
|---|---|---|
| 1 | 0.3393 | 0.3356 |
| 2 | 0.2961 | 0.2954 |
| 3 | 0.3686 | 0.3494 |
| 4 | 0.3673 | 0.3355 |
| 5 | 0.2732 | 0.2774 |
| 6 | 0.3932 | 0.3498 |

*2.8. Construct Validity*

To establish the construct validity of the questionnaire, two different factor analysis approaches were used. The first was the realization of the EFA, and the second, the CFA. The latter was carried out using the structural equation model (SEM). The eigenvalues (quantity of variance in the original set of variables explained by each main component) greater than 1 and the factor load greater than 0.30 were established a priori [73] as criteria acceptable for this study. Statistical software SPSS in its latest version and Amos 24 were used to analyze the data.

## 3. Results

*3.1. Descriptive Statistics*

Descriptive statistics and study of the internal consistency of the Social Skills Scale for Young Immigrants.

Table 6 shows the descriptive statistics of the factors that make up the SSYI: mean, asymmetry, kurtosis, range of scores, and Cronbach's alphas coefficient. The dimensions presented values of asymmetry and kurtosis within the range of normality [74].

The internal consistency indexes ranged between 0.81 and 0.89. This indicates that the instrument has a high level of reliability.

**Table 6.** Descriptive statistics of the dimensions of the Social Skills Scale for Young Immigrants.

| Dimensions | Nº Items | $\alpha$ | Mean | Asymmetry | Kurtosis | Range |
|---|---|---|---|---|---|---|
| To say no | 7 | 0.89 | 6.66 | 0.61 | −0.24 | 12 |
| Self-exp. | 8 | 0.81 | 12.69 | 0.85 | 0.37 | 21 |
| Rights Def. | 4 | 0.81 | 14.17 | 1.09 | 1.57 | 24 |
| Angry Exp. | 3 | 0.81 | 8.15 | 0.74 | −0.01 | 12 |
| Make requests | 4 | 0.85 | 11.00 | −0.25 | 0.60 | 12 |
| Int. Opp. S. | 3 | 0.82 | 6.73 | 0.36 | −0.68 | 9 |
| Scale (SSYI) | 29 | 0.82 | | | | |

Notes: $\alpha$: Internal consistency test, Cronbach's alphas.

### 3.2. Reliability and Validity

After conducting the expert judgment, the instrument was validated using the EFA, a statistical analysis method used to design a model by identifying the correlation between a latent variable and an observed or measured variable [42,75]. This analysis was obtained with the data of 330 immigrants, who answered the 33 items of the questionnaire. This fact ratified the participant-items ratio (10:1) recommended in the literature [76].

Before this analysis, the varimax rotation method was applied to verify that the factors were able to measure if the subjects had social skills [77,78]. The value of Kaiser–Meyer–Olkin (KMO) was 0.81 and the Bartlett Chi-square approximation was 2344.36 with $p = 0.000$. A KMO value close to 1 indicated that the correlation pattern was compact enough to produce different and reliable factors [79]. The results in the Kaiser–Meyer–Olkin and Bartlett sphericity tests indicated that the EFA method was appropriate for use in this study [80] (Table 7).

The statistical program SPSS was used to analyze the data.

**Table 7.** Kaiser–Meyer–Olkin and Bartlett's sphericity tests.

| Kaiser–Meyer–Olkin Measure of Sampling Adequacy | 0.81 |
|---|---|
| Chi-square Approx. | 2344.36 |
| Gl | 528 |
| Bartlett's sphericity test | 0.000 |

According to Siembida et al. [73], after performing the extraction, the communalities must be greater than 0.30 to assume that the measurement has a good validity (Table 8). The analyzed instrument fulfills this criterion in most of the items, except for item number 7 (0.26), number 9 (0.29), number 14 (0.28), and number 16 (0.24), so they were eliminated from the questionnaire, going from having 33 items to the final 29 with which the questionnaire was configured. Further, the EFA study grouped the 33 initial items into six factors or dimensions with rotated factor loads that vary between 0.291 and 0.740 (Table 9). The grouping of the items in the aforementioned dimensions was carried out through the Analysis of Main Components, which is a method of data reduction that allows transforming a set of original variables into a new set with a smaller number of elements [81,82]. To simplify the interpretation of the data obtained in this analysis, in the study, the rotation of the components or factors was carried out through the Varimax orthogonal rotation method, which allows to minimize the number of variables that have high loads in each factor. It is for this reason that Table 6 shows the variables with the highest saturations in each of the six factors obtained.

**Table 8.** Communalities to perform list elimination.

| Items | Extraction |
|---|---|
| 1. I'm afraid they'll laugh at me when I ask questions. | 0.518 |
| 2. It's hard for me to phone other friends. | 0.383 |
| 3. I keep my opinions to myself. | 0.424 |
| 4. I avoid meetings with many people for fear of doing or saying something foolish. | 0.528 |
| 5. It is hard for me to express my feelings to others. | 0.417 |
| 6. If I had to look for a job, I would prefer to write letters/emails rather than having to go through personal interviews. | 0.361 |
| 7. I feel violent when someone of the opposite sex tells me that he likes something about my physical appearance. | 0.267 |
| 8. I find it difficult to express my opinion in groups (in classroom, meetings, etc.) | 0.567 |
| 9. If I buy something and I see that it is not right, I go to the store to return it. | 0.296 |
| 10. When a store first serves someone who came in after me, I do not say anything. | 0.581 |
| 11. If I'm in the cinema and someone bothers me with their conversation, I can hardly tell them to shut up. | 0.377 |
| 12. I am unable to ask for the price of something I am buying to be discounted. | 0.393 |
| 13. When someone "jumps" the queue, I do not say anything. | 0.472 |
| 14. When a friend expresses an opinion with which I do not agree, I do not say anything, even if I think otherwise. | 0.284 |
| 15. When a close relative bothers me, I prefer to hide my feelings rather than expressing anger. | 0.405 |
| 16. I find it hard to express aggression or anger toward the opposite sex, even if I have justified reasons. | 0.241 |
| 17. I prefer to shut up to avoid problems with other people. | 0.612 |
| 18. If a seller insists on showing me a product that I do not want, it's hard for me to say I do not want it. | 0.420 |
| 19. When I'm in a hurry and a friend calls me on the phone, it's hard for me to hang up. | 0.389 |
| 20. When someone borrows my things from me, I lend them, even if I do not want to or I do not like it. I do not know how to say no. | 0.383 |
| 21. I do not know how to tell a friend that he talks a lot, that he stops talking. | 0.459 |
| 22. When I decide that I do not want to go out with someone again, it's hard for me to tell it. | 0.419 |
| 23. When someone calls me to leave, I do not know how to refuse, even though I do not feel like it. | 0.520 |
| 24. I find it difficult to ask for something to be returned. | 0.480 |
| 25. If, in a restaurant, they do not bring me the food as I had requested, I call the waitress and ask the cook to do it again, | 0.470 |
| 26. If I leave a store and I realize that they have given me the change badly, I return there to ask for the correct change, | 0.529 |
| 27. If I lend money to a friend and he does not return it, I can remember it, | 0.434 |
| 28. It's hard for me to ask favors from my friends, | 0.343 |
| 29. When I like a boy or a girl I know what to say, | 0.523 |
| 30. When I have to flatter someone, I do not know what to say, | 0.356 |
| 31. I prefer to keep quiet so as not to create problems for other people, | 0.609 |
| 32. If I find a person I like, I approach him to talk, | 0.382 |
| 33. I am not able to ask someone for an appointment, | 0.379 |

In addition, the principals' components analysis showed that the six dimensions have 43.09% of the total variance explained.

The distribution of the items in six factors coincides with the hierarchical structure of the reference instrument [1], which bases the dimensions of social skills in important theoretical references [83–88], which specify that when a subject presents social skills, he is able to express his emotions in social situations, defend his own rights, show denial and make requests, as well as initiate interactions with people of another gender. These are aspects that are developed in the six dimensions established in the EFA (the ability to say no and cut interactions, self-expression in social situations, the defense of one's rights as a consumer, the expression of anger or disagreement, the ability to make requests to others, and the ability to initiate positive interactions with people of the opposite sex).

Finally, CFA was carried out, through which it was intended to configure the final instrument (Figure 1). This second-order analysis validated the hierarchical structure of the instrument [89–91] and checked the previously defined relationships in the EFA between the variables. In addition, it is considered effective to determine the reliability [92] and to measure the invariance of the measurement between groups, thus checking that the instrument is appropriate to examine the equivalence of measurement with respect to two particularly relevant variables in relation to the construct evaluated [93].

The CFA was carried out in the investigation to complement the results obtained with the EFA. It is necessary to perform this second-order analysis to ratify and complement any doubt about the validity of a questionnaire [94], as it complements the factor analysis carried out [95].

To carry out this analysis, the questionnaire was administered again to young immigrants from the cities of Granada, Málaga, Almería, Jaén, Córdoba, Sevilla, Cádiz, Huelva, Ceuta, and Melilla, in this case to a sample of 568 participants. The analysis was carried out using the AMOS software in version 24.

**Table 9.** Grouping of the 33 items into six factors with rotated factor loads.

| Items | Factors | | | | | |
|---|---|---|---|---|---|---|
| | 1 | 2 | 3 | 4 | 5 | 6 |
| 23 | 0.639 | | | | | |
| 21 | 0.635 | | | | | |
| 24 | 0.599 | | | | | |
| 20 | 0.595 | | | | | |
| 22 | 0.573 | | | | | |
| 19 | 0.530 | | | | | |
| 18 | 0.468 | | | | | |
| 14 | 0.291 | | | | | |
| 4 | | 0.713 | | | | |
| 1 | | 0.658 | | | | |
| 8 | | 0.651 | | | | |
| 5 | | 0.539 | | | | |
| 3 | | 0.529 | | | | |
| 2 | | 0.494 | | | | |
| 6 | | 0.413 | | | | |
| 7 | | 0.389 | | | | |
| 28 | | 0.366 | | | | |
| 10 | | | 0.716 | | | |
| 13 | | | 0.653 | | | |
| 11 | | | 0.464 | | | |
| 12 | | | 0.454 | | | |
| 17 | | | | 0.740 | | |
| 31 | | | | 0.731 | | |
| 16 | | | | 0.385 | | |
| 26 | | | | | 0.682 | |
| 25 | | | | | 0.618 | |
| 27 | | | | | 0.610 | |
| 9 | | | | | 0.504 | |
| 29 | | | | | | 0.571 |
| 33 | | | | | | 0.529 |
| 15 | | | | 0.297 | | |
| 32 | | | | | 0.395 | |
| 30 | | | | | | 0.393 |

The CFA is presented in route diagrams where the circles represent latent variables and the squares represent observed variables [96]. The single-headed arrows are used to imply an assumed direction of influence, and the two-headed arrows represent the covariance between the six latent variables [97] (Figure 1).

In this analysis, the value of CMIN or Chi-square adjustment test ($\chi^2$) of Pearson was measured, which was 765.98 with $p$ = 0.000, being statistically significant [92]. However, the Chi-square value shows great sensitivity to the sample size [98], which is why we decided to use other adjustment indexes in order to contrast the model [99]: the index of comparative adjustment (CFI), the Tucker–Lewis index (TLI), and the root mean square error of approximation (RMSEA) are the most relevant [99,100]. The CFI and the TLI have a range of 0 to 1 considering these values more valid when they are closer to the unit [101]; in addition, the value of RMSEA is considered to indicate a good fit to the model if it is less than 0.06 [102,103]. In the study, the CFI was $n$ = 0.877, the TLI was $n$ = 0.852 (Table 10). and the RMSEA that was obtained was 0.04 (Table 11), data that show a very good fit to the hypothetical model. The method of maximum verisimilitude was chosen to carry out this analysis [104].

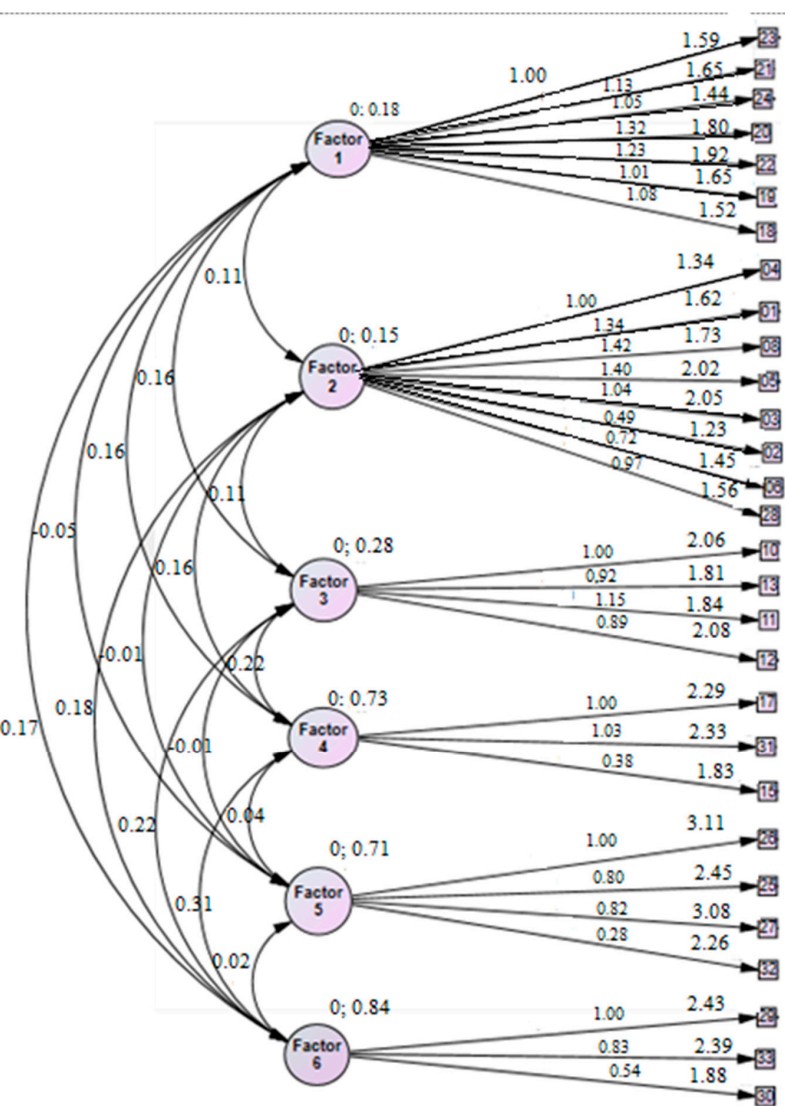

**Figure 1.** Confirmatory factor analysis diagram.

**Table 10.** Fit test of the comparative index and the Tucker–Lewis index.

| Model | Tucker–Lewis Index | Index of Comparative Adjustment |
|---|---|---|
| Default model | 0.85 | 0.88 |
| Saturated model | | 1 |
| Independence model | 0.000 | 0.000 |

**Table 11.** Mean squared error approximation test.

| Model | Square Error of Approximation |
|---|---|
| Default model | 0.04 |
| Independence model | 0.11 |

Finally, the latent variables for the subscales of social skills had seven, eight, four, three, four, and three items (Figure 1). The results of the route diagram showed acceptable factor loads (standardized values) for all the items, ranging from 0.28 to 1.42, these values being acceptable if they are above 0.10 [105]. The factorial loads of seven elements of the construct of the ability to say no and cut interactions was between 1 and 1.32. The eight elements of the construct self-expression in social

situations was between 0.49 and 1.42. The four elements of the construct defense of one's rights as a consumer was between 0.89 and 1.15. The three elements of the construct expressing anger or disagreement was between 0.38 and 1.03. The four elements of the construct making requests was between 0.28 and 1, and, finally, the three elements of the construct initiating interactions with the opposite sex was between 0.54 and 1.

Once the CFA was finished and the instrument was confirmed in 29 items (Appendix A), the internal consistency test (Cronbach's Alpha) was re-run, in which on this occasion a value of $\alpha = 0.82$ was established, A more acceptable value of reliability than the one previously obtained when the instrument consisted of 33 items.

## 4. Discussion and Conclusions

The objective of this study was to develop an instrument to assess the social skills of young immigrants from different Spanish cities.

The internal consistency and test–retest reliability of the subscales were considered acceptable and are very close to those found in previous studies with reference instruments [2,26–28,34,40–42]. More specifically, taking into account the data obtained in the total reliability of $\alpha = 0.82$ and in the reliability for each of the dimensions, from $\alpha = 0.81$ to $\alpha = 0.89$, it is considered to be a highly reliable instrument [81,95,97]. This aspect beats the reliability properties of other scales that measure social skills such as the Social Skills Questionnaire (CHASO III) [2] with $\alpha = 0.70$, the Social Skills Questionnaire for College Students (SSQ-U) [29] with $\alpha = 0.70$, or the Social Skills Questionnaire for Traumatic Brain Injury (SSQ-TBI) [27] with $\alpha = 0.80$.

This means that the questionnaire can measure the social skills possessed by young immigrants, and, therefore, researchers can rely on their use.

In addition, the apparent, criterion, and nomological validity, as well as the readability analysis of the subscales, were also satisfactory. These aspects are very important, since the objective of the research was to make an easily understandable instrument that measured relevant dimensions for the diagnosis of social skills in subjects with linguistic limitations because they are in the process of learning the host language. This fact, together with the reduction of factors that began to be evident in the content validity process, increases the value of the instrument. It is preferred to build instruments that are easy to complete and quick to make [59,106], which stands out against the longer scales existing in previous studies, among which are: Social Skills Questionnaire (CHASO III) [2] with 76 items, Social Skills Inventory [34] with 90 elements, Social Skills Questionnaire for College Students (SSQ-U) [33] with 38 items, the Social Skills Improvement System-RS [26] with 46 items, and the Social Skills Questionnaire for Traumatic Brain Injury (SSQ-TBI) [27] with 41 elements.

It should be taken into account that the sample to which the scale is directed consists of young people between 12 and 18 years of age of different ethnicities, races, or religions who are adapting to the language and customs of the host country. That is why this instrument has taken into account, when preparing the elements of the questionnaire, to avoid value judgments or statements that may offend the various members to whom it may be applied. That is why, for the preparation of the questionnaire, an inclusive language is used, defining this as the one that is used in a neutral way with the intention that people belonging to minority groups do not feel apart [107,108].

With respect to the validation of constructs, the analysis of the SEM results indicated that, in general, the model was adequate.

The Exploratory Factor Analysis made it possible to select those items that presented a better psychometric behavior, going from a scale of 33 items to a scale of 29, with six different dimensions, which presented adequate levels of correlation with each other, indicative of an excellent validity, which was confirmed through the Confirmatory Factor Analysis. As it is a validation of the adaptation of a questionnaire, it could be expected that the results in terms of the indices would be lower than that of the starting instrument [1], but the reality is that it surpasses it, having better psychometric properties of validity and reliability [104], in addition to improving statistical power by reducing the

length of the questionnaire, maintaining its original structure and functioning, but with the advantage of needing fewer items to carry out the study [109]. Even so, the analysis of the construct validity of the SSYI was shown in consonance with the literature [1], since six dimensions were also found. In this sense, the dimensions of the resulting scale encompass all possible levels of social skills.

The evaluation of the model of structural equations verified that the derived coefficients have a positive direction according to the theory [101]. In summary, the results obtained show a good reliable fit, which allows it to be a candidate perfect for assessing the social skills of young immigrants, becoming the only instrument that takes into account the linguistic and comprehension peculiarities of the sample to which it is addressed.

## 5. Limitations

The main limitation comes from the number of participants in relation to the sample size. Futures research studies should try to expand the sample, although it is important to highlight the difficulty to work with this type of participants, as well as to consider the sample presented in this study as sufficient for validation of a questionnaire.

On the other hand, the use of self-reports with a small number of items allows a series of advantages over other evaluation methods. It was applied quickly and inexpensively, in addition to its easy correction and interpretation; however, it also has some limitations such as the answers randomly that participants can perform and the difficulty that some immigrants may have to inform about their own thoughts, states, behaviors, cognitions, or affects, among other things [4,41,75].

In addition, the instrument can find limitations related to the understanding of the language, since the majority of young people come from countries whose mother tongue differs from Spanish [3]. That is why, in the application of the questionnaire, these immigrants require the help of a social worker who dominates their original language, as has been shown in this investigation.

## 6. Implications for Practice

On the basis of this study, it can be deduced that the current scale, although new, may be useful for researchers and social workers who move in the field of social skills and immigration. Researchers can use the instrument to predict whether young immigrants have the necessary social skills for a correct integration in the society and, based on their results, they can design intervention and inclusion programs for them. In such interventions, the constructs of the SSYI can be measured before and after the actions to evaluate the changes in these behavioral precursors.

Another advantage of this instrument is that it also operates acculturation, which is an important determinant for the immigrant population, because they must assimilate the interaction with the native subjects in the destination culture [110]. Therefore, the instrument also serves to predict socially appropriate or inadequate skills in host immigrants.

Finally, the interventions carried out by social workers should aim to integrate the young immigrants, taking into account the construction of the dimensions to which the SSYI is adjusted. In this way, the ability to say no and cut interactions can be constructed by discussing the negative consequences that can happen when one does not know how to say no and ends up doing something one does not want to do; the ability to self-express in social situations can be built by holding debates where young immigrants are given guidelines to follow for a correct expression of their feelings; the defense of one's rights as a consumer can be improved by using role-playing, where young immigrants put themselves in the shoes of a buyer, thus describing the beneficial effects of knowing what their rights are; the ability to express anger or disagreement can be worked out when a conflict occurs and a solution is sought, causing immigrants to express their disagreement calmly and coherently; the capacity to make requests can be carried out in different contexts, making the young immigrant interact with people of different cultural, ethnic, or religious origins; and finally, the ability to initiate positive interactions with people of the opposite sex can be constructed by dividing desirable behaviors into small steps, involving role models, using persuasion and reinforcement, and reducing stress [110–114].

**Author Contributions:** Conceptualization, C.F.-L. and M.T.-F.; methodology, C.F.-L. and M.T.-F.; software, C.F.-L.; validation, C.F.-L., formal analysis, C.F.-L. and M.T.-F.; investigation, C.F.-L.; resources, C.F.-L.; data curation, C.F.-L.; writing—original draft preparation, C.F.-L. and M.T.-F.; writing—review and editing, C.F.-L., M.T.-F. and E.M.O.-M.; visualization, M.T.-F.; supervision, M.T.-F. and E.M.O.-M.; project administration, E.M.O.-M.; funding acquisition, E.M.O.-M. All authors have read and agreed to the published version of the manuscript.

**Funding:** This work was funded by the Research Project Competitive EDU2017-88641-R. (INTEREDUC/UFM): "Hybrid learning models for educational intervention with unaccompanied foreign minors (UFM). Effective tools for the minor's school and social integration."

**Conflicts of Interest:** The authors declare no conflict of interest.

## Appendix A

Social Skills Scale for Young Immigrants (SSYI)

Age:

Gender:

Province:

Nationality:

Instructions:

Mark with an X the answer that you believe that is more according to your personality:

1. It never happens to me.
2. Sometimes it happens to me.
3. It happens to me a lot.
4. It always happens to me.

You remember this answer is anonymous.

|  |  | Never 1 | Sometimes 2 | A Lot 3 | Always 4 |
|---|---|---|---|---|---|
| 1 | I am afraid they will laugh at me when I ask questions. |  |  |  |  |
| 2 | I am not able to phone other friends. |  |  |  |  |
| 3 | I keep my opinions to myself. |  |  |  |  |
| 4 | I do not like meeting with many people for fear of doing or saying something foolish. |  |  |  |  |
| 5 | I am not able to express my feelings to others. |  |  |  |  |
| 6 | If I had to look for a job, I would prefer to write letters/emails rather than having to go through personal interviews. |  |  |  |  |
| 7 | I find it difficult to express my opinion in groups (in classroom, meetings, etc.). |  |  |  |  |
| 8 | When in a store first serves someone, who came in after me, I do not say anything. |  |  |  |  |
| 9 | If I am in the cinema and someone bothers me with their conversation, I cannot tell him/her to shut up. |  |  |  |  |
| 10 | I am unable to ask for the price of something I am buying to be discounted. |  |  |  |  |
| 11 | When someone "jumps" the queue, I do not say anything. |  |  |  |  |
| 12 | When a close relative bothers me, I prefer to hide my feelings rather than express anger. |  |  |  |  |
| 13 | I prefer to shut up to avoid problems with other people. |  |  |  |  |
| 14 | If a seller insists on showing me a product that I do not want, it is hard for me to say I do not want it. |  |  |  |  |
| 15 | When I am in a hurry and a friend calls me on the phone, I do not know how stop the conversation and hang up. |  |  |  |  |
| 16 | When someone borrows my things from me, I lend them, even if I do not want to do. I do not know how to say no. |  |  |  |  |
| 17 | I do not know how to tell a friend that he talks a lot, that he stops talking. |  |  |  |  |
| 18 | When I do not want to go out with someone again, I cannot tell it to him/her. |  |  |  |  |
| 19 | When someone calls me to leave, I do not know how to refuse, even though I do not feel like it. |  |  |  |  |
| 20 | I do not know how to ask someone to give me back something that I lent him/her. |  |  |  |  |
| 21 | If, in a restaurant, they do not bring me the food as I had requested, I cannot call the waitress or ask the cook to do it again. |  |  |  |  |
| 22 | If I leave a store and I realize that they have given me the change badly, I do not return there to ask for the correct change. |  |  |  |  |
| 23 | If I lend money to a friend and he does not return it, I cannot remember it to them. |  |  |  |  |

| | | Never 1 | Sometimes 2 | A Lot 3 | Always 4 |
|---|---|---|---|---|---|
| 24 | I am not able to ask favors from my friends. | | | | |
| 25 | When I like a boy or a girl, I do not know what to say. | | | | |
| 26 | When I have to flatter someone, I do not know what to say. | | | | |
| 27 | I prefer to keep quiet so as not to create problems for other people. | | | | |
| 28 | If I find a person I like, I cannot approach him to talk. | | | | |
| 29 | I am not able to ask someone for an appointment. | | | | |

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
