# Peer review of "Exploratory and Confirmatory Factor Analysis of the Social Skills Scale for Young Immigrants"

_sustainability, doi:10.3390/su12176897_

Round 1

Reviewer 1 Report

The article presented is very interesting for the scientific community,
the theoretical framework is situated in 2020, well constructed and updated.
The authors presented are relevant to the research.
The empirical framework is well constructed, the scale is well explained, well built and expressed in the article.
Correlation analysis through Spearman's Rho is correct.
The confirmatory factor analysis is correct and well constructed, it is
a good article.

Author Response

Reviewer 1 has not indicated anything to change.

Reviewer 2 Report

Abstract and introduction should specify that Spain is the host country being studied and that these are young Spanish immigrants. Paper could also benefit from a more detailed description of the unique challenges Spanish immigrants face in their host country. Currently the introduction is too broad. For example, many young Chinese immigrants in America do not face the same set of circumstances, particularly if they come from wealthy SES. As such, it's important to distinguish the population and host country characteristics for the reader.

Author Response

Comment 1

Abstract and introduction should specify that Spain is the host country being studied and that these are young Spanish immigrants.

AUTHOR RESPONSE

Following the recommendation of the reviewer, in line 17 of the summary it has been incorporated that these are young immigrants whose host country is Spain.

Comment 2

Paper could also benefit from a more detailed description of the unique challenges Spanish immigrants face in their host country. Currently the introduction is too broad. For example, many young Chinese immigrants in America do not face the same set of circumstances, particularly if they come from wealthy SES. As such, it's important to distinguish the population and host country characteristics for the reader.

AUTHOR RESPONSE

Taking into account the reviewer's comment, from lines 38 to 56, a brief description of the challenges faced by young immigrants when arriving in Spain has been made. Distinguishing the country of origin of the same and the characteristics of the host country in the reception process.